# GENERATIVE ADVERSARIAL NETWORK-BASED VIRTUAL TRY-ON WITH CLOTHING REGION

**Shizuma Kubo, Yusuke Iwasawa, and Yutaka Matsuo**
The University of Tokyo
Bunkyo-ku, Japan
`{kubo, iwasawa, matsuo}@weblab.t.u-tokyo.ac.jp`

## ABSTRACT

We propose a virtual try-on method based on generative adversarial networks (GANs). By considering clothing regions, this method enables us to reflect the pattern of clothes better than Conditional Analogy GAN (CAGAN), an existing virtual try-on method based on GANs. Our method first obtains the clothing region on a person by using a human parsing model learned with a large-scale dataset. Next, using the acquired region, the clothing part is removed from a human image. A desired clothing image is added to the blank area. The network learns how to apply new clothing to the area of people's clothing. Results demonstrate the possibility of reflecting a clothing pattern. Furthermore, an image of the clothes that the person is originally wearing becomes unnecessary during testing. In experiments, we generate images using images gathered from Zaland (a fashion E-commerce site).

## 1 INTRODUCTION

In recent years, demand for EC (E-commerce) sites in fashion field has been increasing. It is expected that the fashion EC market will continue to expand. When choosing clothes, it is important that they are suitable to a person, and that they can be checked and tried on at a store. However, such trials are difficult at fashion EC sites. If some way exists to judge whether clothes are appropriate for a person, then there is a high probability that the purchasing experience at the fashion EC site will be improved. As a technology trend, research into generative adversarial networks (GANs) (Goodfellow et al., 2014), which is a deep generative model, is progressing. It is effective for the generation of images. Particularly, Conditional Analogy GAN (CAGAN) (Jetchev & Bergmann, 2017) has already been proposed as research on dressing of people on the image. Because such a virtual try-on can be a key to judging whether clothes will suit a person, it is possible to improve customer purchasing experiences at the EC site if the accuracy of the method improves.

In this work, we intend to improve the virtual try-on architecture for the complex texture pattern, for which we propose and evaluate a network that explicitly incorporates human parsing model so that it is effective for complex patterns of clothing.

## 2 METHOD

### 2.1 PROPOSED NETWORK

The process of the proposed method is similar to that portrayed in Figure 1. In other words, by inputting the image of the person and the image of the clothes to be worned, the dressed human image is generated. In this proposed method, dressing with a upper-body garments is done. The conventional method (CAGAN) uses two networks: a Generator that carries out dressing change and a Discriminator that judges whether the dress change is done properly. The proposed method has these two networks. Discriminator has the same structure as CAGAN. However, we incorporated a human parsing network into Generator to examine the clothing area.

First, human parsing is performed to specify the region of the clothing of the inputted person image. This human parsing network uses the model proposed by Gong et al. (2017). The network

parameters of this model are not updated in the learning of this proposed method but use the learned parameters by the LIP dataset proposed in the same paper. Subsequently, using the image obtained using the human parsing network as input of Encoder-Decoder's network, an image of the dressed person is generated. An image obtained by removing the region of clothing from the image of a person is also input. The EncoderDecoder network is based on the network of Isola et al. (2017).

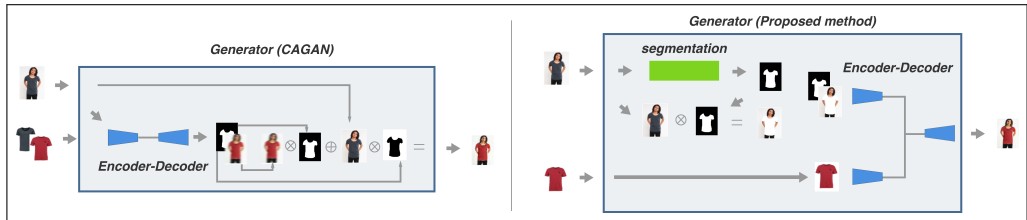

Figure 1: Schematic of CAGAN's Generator (shown on the left side) and the proposed method's Generator (shown on the right side). First, the proposed method's Generator detects the region of clothing of the input person's image (segmentation). Then prepares the personal image from which the region is removed. Then Encoder-Decoder generates a dress up image.

## 2.2 LOSS FUNCTION

The loss function of the proposed method is defined as Equation (1). Here, G and D respectively represent networks of Generator and Discriminator networks. Generator and Discriminator learn alternately to maximize and minimize V (D, G), respectively. The loss function is largely divided into three terms, each of which is described below.

$$\min_G \max_D V(D,G) = L_{cGAN}(G,D) + L_{cyc}(G) + L_{perceptual}(G). \tag{1}$$

First is a term related to hostile learning of Discriminator and Generator defined by Equation (2). $x_i$ represents the image of the person and $y_i$ represents the image of the clothes the person is wearing. The dataset comprises a set of $N$ pairs such as $\{x_i, y_i\}_{i=1}^N$.

$$
\begin{aligned}
L_{cGAN}(D,G) = & E_{x_i,y_i \sim p_{data}}[log D(x_i, y_i)] \\
& + E_{x_i,y_j \sim p_{data}}[log(1 - D(G(x_i, y_j), y_j))] \\
& + E_{x_i,y_j \sim p_{data}}[log(1 - D(x_i, y_j))].
\end{aligned}
\tag{2}
$$

Second is a term that expects to return to the original image when clothes are changed once. The clothes are changed to the original clothes once again, as defined by equation (3), similarly to Zhu et al. (2017). This term takes the L1 loss between the generated image and the original image.

$$L_{cyc}(G) = E_{x_i,y_i,y_j \sim p_{data}}||x_i - G(G(x_i, y_j), y_i)||. \tag{3}$$

Finally, it is a term called perceptual loss defined by equation (4), similar to that described by Johnson et al. (2016). It is expected that the output obtained by inputting the pair $(x_i, y_i)$ into the Generator and the original person image $x_i$ will be the same. This term consists of the sum of difference $l_\phi$ of the feature map of each block obtained by inputting each image into the learned model of VGG19 (Simonyan & Zisserman, 2015), which showed high performance in general object recognition. $\lambda$ is the reciprocal of the number of parameters of each layer.

$$L_{perceptual}(G) = E_{x_i,y_i \sim p_{data}}\left[ \sum_{i=1} \lambda_i l_{\phi,blocki\_conv2} \right]. \tag{4}$$

## 3 EXPERIMENTAL RESULTS

### 3.1 SETTING

A dataset of many pairs of a person image and an image of clothes worn by the person for learning was acquired from the website of the Zaland[1] fashion EC site. The image size was $128 \times 96$ pixels. The person in the image was limited to facing forward. The clothing image was limited to that showing only one piece of clothing. Others were removed as noise. Of the 9286 total pairs, 9000 pairs were used for learning; 286 pairs were used for testing.

Implementation was done using Keras, a typical DeepLearning framework, at the Tensorflow back-end. Adam (Kingma & Ba, 2015) was used as an optimization method. In addition, the network is formed by stacking the convolution layers and deconvolution layers in multiple layers. Each layer performs batch normalization and uses ReLU or Leaky ReLU as an activation function. Similarly, CAGAN was also implemented and compared.

### 3.2 RESULTS

Several indicators for evaluating models that generate images are proposed such as the Inception Score (Salimans et al., 2016) and FID (Heusel et al., 2017), but they are not useful for judging the quality of dressing. Therefore, the effectiveness of the proposed method was evaluated by question-naire. For the test, 30 dressing changes were made, respectively, using images in the test dataset with the proposed method and CAGAN. Then a questionnaire was made as to which one was appropriate as a dress change. The answer was obtained from 131 people. By averaging all questions, the share of respondents answering that the proposed method was appropriate was 82.2%, indicating the effectiveness of the proposed method.

Figure 2 presents an example of a comparison of generated results. By considering the clothing region, it is thought that the influence of the pattern of the original clothing becomes small. The reflection of the pattern of the clothes to be dressed is well done.

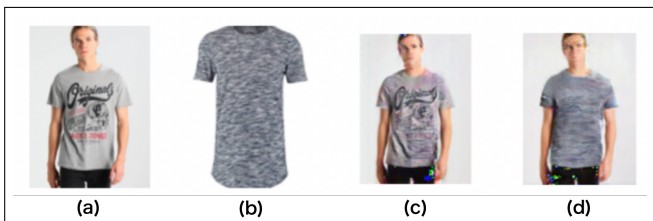

Figure 2: (a) and (b) are dataset images. Change the clothes of person (a) to clothes of (b). (c) shows the result of CAGAN. (d) is the result obtained using the proposed method.

## 4 CONCLUSION

We proposed and evaluated a virtual try-on method based on GANs by considering the region of clothes. Its effectiveness is shown in comparison with CAGAN. Furthermore, the image of clothes worn by the person is unnecessary in testing for the proposed method. However, it is necessary for CAGAN.

Even in this paper, some room for improvement remains in relation to the pattern of clothes. Some limitations exist on the scope of applications such as restricting the posture of a person or limiting clothes to the upper body. Further improvement is anticipated in this respect.

---

[1]zaland: https://www.zalando.de

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

APPENDIX - QUALITATIVE COMPARISONS OF DIFFERENT METHODS

Figure 3 presents several examples of a comparison of generated results. As you can see from the first row of the figure, in the case of simple clothing, the results of CAGAN and the proposed method are not so different. However, in other cases the proposed method looks better than CAGAN.

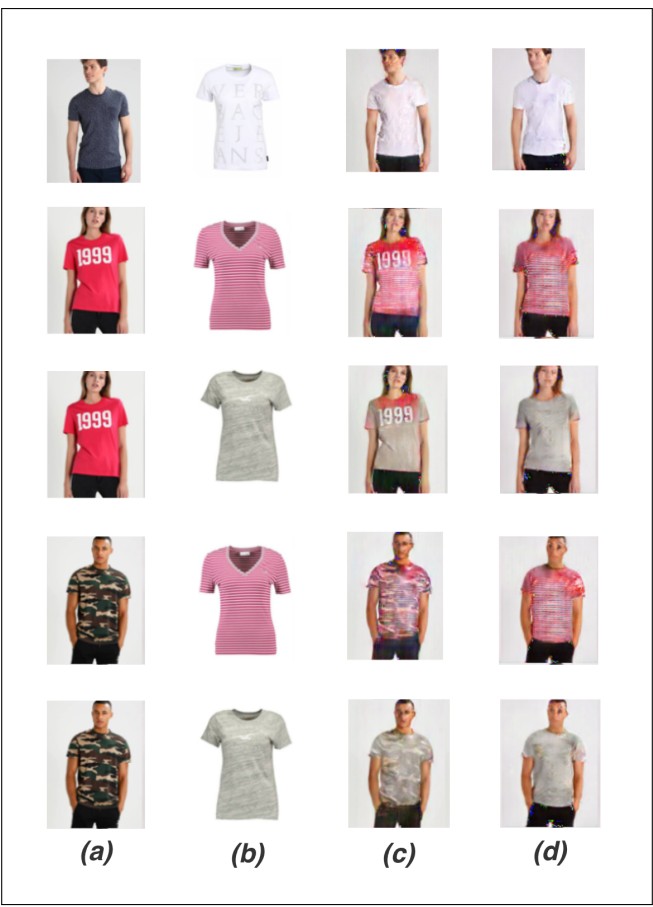

Figure 3: (a) and (b) are dataset images. Change the clothes of person (a) to clothes of (b). (c) shows the result of CAGAN. (d) is the result obtained using the proposed method.

