# OpenReview forum: "Generative Adversarial Network-Based Virtual Try-On with Clothing Region"
_ICLR.cc/2018/Workshop — Reject_

### Official Review · AnonReviewer3 · 2018-03-09
**Lacking experiments**

**Rating:** 4
**Confidence:** 3

**Review:**

This paper proposes a system that changes the dressing of a person in an image, without the original dressing image. This is done via segmentation provided by human parsing network. This is an improvement over prior works (CAGAN). However, there is no much experiments in the paper, except for a user study that asks whether the proposed method is more appropriate than CAGAN, and a few examples.

Overall, more experiments, in particular quantitative experiments, are needed to justify the method.

Note that I am not an expert in GAN so my ideas might be wrong.

---

### Official Review · AnonReviewer1 · 2018-03-10
**Incremental work for unsatisfying results**

**Rating:** 3
**Confidence:** 5

**Review:**

The GAN approach has been compared to CAGAN for inpainting different clothes to an input image of a person in the foreground. The approach amounts to changing the loss function injecting some new terms, the newer one is a perceptual base one.

The results are not satisfying, even if better than CAGAN. In fact, the new clothes are remarkably changed (the stripes red t-shirts becomes a washed out garment). Unfortunately, these are the details that let a person choose and buy an item. Plus, the face of the persons become distorted. Finally, candidate people having diverse types of body shape (endomorph, mesomorph, ectomorph)  should have been included

---

### Official Review · AnonReviewer2 · 2018-03-11

**Rating:** 5
**Confidence:** 4

**Review:**

In this paper the authors consider a modification of the original approach of CAGAN for generating virtual try on. The main contribution is to include an explicit segmentation term.

A drawback of the current approach is that we do not see any baseline comparisons with the original network in terms of some quantitative metric say using an independent Wasserstein critic or by using a user evaluation for the same.

There has also been other significant work done in this space such as "Fashion Style Generator", IJCAI 17 and VITON, a image based virtual try on network. However, as no comparisons are provided it is not possible to obtain comparisons with the other works.

---

### Decision · Program_Chairs · 2018-03-20
**ICLR 2018 Workshop Acceptance Decision**

**Decision:**

Reject

**Comment:**

Based on the reviews, this paper has not been accepted for presentation at the ICLR workshop. However, the conversation and updates can continue to appear here on OpenReview.